# Genetic Inactivation of Peroxiredoxin-I Impairs the Growth of Human Pancreatic Cancer Cells

**DOI:** 10.3390/antiox10040570

**Published:** 2021-04-08

**Authors:** Hajar Dahou, Marie-Albane Minati, Patrick Jacquemin, Mohamad Assi

**Affiliations:** De Duve Institute, Université Catholique de Louvain, 1200 Brussels, Belgium; hajar.dahou@uclouvain.be (H.D.); marie-albane.minati@uclouvain.be (M.-A.M.)

**Keywords:** antioxidants, autophagy, cell cycle, pancreatic cancer, ROS

## Abstract

Pancreatic ductal adenocarcinoma (PDAC) is an aggressive disease with few therapeutic options. The identification of new promising targets is, therefore, an urgent need. Using available transcriptomic datasets, we first found that *Peroxiredoxin-1* gene (*PRDX1*) expression was significantly increased in human pancreatic tumors, but not in the other gastrointestinal cancers; its high expression correlated with shortened patient survival. We confirmed by immunostaining on mouse pancreata the increased Peroxiredoxin-I protein (PRX-I) expression in pancreatic neoplastic lesions and PDAC. To question the role of PRX-I in pancreatic cancer, we genetically inactivated its expression in multiple human PDAC cell lines, using siRNA and CRISPR/Cas9. In both strategies, PRX-I ablation led to reduced survival of PDAC cells. This was mainly due to an increase in the production of reactive oxygen species (ROS), accumulation of oxidative DNA damage (i.e., 8-oxoguanine), and cell cycle blockade at G2/M. Finally, we found that PRX-I ablation disrupts the autophagic flux in PDAC cells, which is essential for their survival. This proof-of-concept study supports a pro-oncogenic role for PRX-I in PDAC.

## 1. Introduction

Pancreatic ductal adenocarcinoma (PDAC) remains one of the deadliest cancers in the world, with few, if any, efficient treatments. The 5-year survival rate for PDAC is 6–9%, and its incidence (+77%) and mortality (+79%) are continuously increasing [1], highlighting the unmet need for new therapeutic strategies. During the past decade, genetically engineered mouse models (GEMM), phenocopying human malignancies, unveiled mechanisms underlying PDAC development [2,3]. In the presence of *Kirsten rat sarcoma* (*Kras)* mutations and pancreatitis (inflammation of the pancreas), pancreatic acinar cells undergo acinar-to-ductal metaplasia (ADM), eventually leading to the formation of pancreatic intraepithelial neoplasia (PanIN) and PDAC [4,5]. Cholecystokinin plays a key role in driving PDAC by promoting pancreas inflammation [6,7]; cerulein, a cholecystokinin analog, is frequently used to induce experimental pancreatitis and accelerate PDAC development [4]. In addition, cholecystokinin and other hormones such as secretin shape pancreas function by regulating endocrine and nervous pathways [8].

At the molecular level, emerging research revealed the presence of redox-sensitive mechanisms essential for the maintenance of PDAC growth [9,10]. These include the modulation of reactive oxygen species (ROS) levels, and the control of the expression and activity of the redox master regulator, nuclear factor erythroid 2-related factor 2 (NRF2), as well as other important antioxidant enzymes [9,10]. On this basis, Peroxiredoxins (PRXs) constitute an important checkpoint for tumor development and growth [11], as they regulate the intracellular levels of various ROS (i.e., hydrogen peroxide) and influence the activation of pro-oncogenic signaling cascades, such as Extracellular signal-regulated kinases/mitogen-activated protein kinases (ERK/MAPK) and Janus kinase/signal transducer and activator of transcription 3 (JAK/STAT3) [12].

Typical 2-cysteines (2-Cys) PRX-I, -II, -III, and -IV are homodimeric enzymes that possess a high catalytic efficiency (10^7^ M^−1^ s^−1^) at the peroxidatic cysteine site [13]; their implication in the pathophysiology of cancer has been previously reported [12]. Yet, the role of PRX-I in cancer development is still a subject of debate. Cell culture and rodent models have shown that PRX-I promotes the development of lung cancer in the presence of the oncogenic protein Kirsten rat sarcoma (KRAS^G12D^) [14], whereas it represses Harvey rat sarcoma viral oncogene homolog (HRAS^G12V^)-driven liver tumor growth [15]. Therefore, the role of PRX-I appears to be organ- and context-specific. In the pancreas, the role of PRX-I is not clearly elucidated. However, our results and that of others suggest a pro-oncogenic role for PRX-I in PDAC [16,17]. In support of these previous results, we found that *Peroxiredoxin-1* mRNA was significantly upregulated in human pancreatic tumors and that its high expression correlated with shortened patient survival (Figure 1A,B). To study the functional role of PRX-I, we used molecular and redox biology techniques on human PDAC cell lines. Our present results support a pro-oncogenic role for PRX-I in PDAC.

## 2. Materials and Methods

### 2.1. Mice and Ethical Approval

All procedures described were approved by the animal welfare committee of the University of Louvain Medical School (ethic number: 2017/UCL/MD/020). Three *ElastaseCreER-LSLKras^G12D^-p53^R172H^* (*KPE*) male mice of 30 to 35 g (total body weight) were maintained in a CD1-enriched background. *ElaCreER*, *LSL-Kras^G12D^*, and *p53^R172H^* mice have been described previously [3,18,19]; these mice were bred in our animal facility to obtain a *KPE* background. Adult *KPE* mice (6-weeks-old) were first treated with tamoxifen (30 mg/mL in corn oil) by oral gavage (T5648-1G, Sigma Aldrich, Overijse, Belgium) and 4-hydroxytamoxifen (0.3 mg/mL in corn oil) by subcutaneous injection (H7904-25 mg, Sigma Aldrich, Overijse, Belgium) to recombine the LSL stop cassettes and allow the expression of *Kras^G12D^* and *p53^R172H^* from their respective endogenous locus. Next, mice received intraperitoneal injections of cerulein (100–150 µL in phosphate-buffered saline, PBS) (125 µg/Kg; 24252, Eurogentec, Seraing, Belgium) for 6 weeks to induce pancreas inflammation, and were then kept for 14 weeks to allow tumor development. Detailed information regarding the used tamoxifen and cerulein regimens can be found in Appendix A. Mice were sacrificed at 28-weeks-old by cervical dislocation, and pancreata were collected to analyze PRX-I expression by immunohistochemistry. Three untreated age- and sex-matched *wild-type* mice (without cerulein treatment) were used as controls to assess the basal expression of PRX-I in the normal pancreas. 

### 2.2. Cell Lines, Small Interfering RNA (siRNA) Transfections and Pharmacological Treatments

PANC-1 cells (CRL-1469, ATCC, Manassas, VA, USA) were maintained in Dulbecco’s modified eagle medium (DMEM) (61965026, Life Technologies, Merelbeke, Belgium) supplemented with 10% fetal bovine serum (FBS) (F9665-500ML, Sigma Aldrich, Overijse, Belgium), 1% penicillin-streptomycin (15070063, Life Technologies, Merelbeke, Belgium) and 1% sodium pyruvate (11360070, Life Technologies, Merelbeke, Belgium). MIAPaca-2 cells (CRL-1420, ATCC, Manassas, VA, USA) were maintained in DMEM supplemented with 10% FBS, 1% penicillin-streptomycin, and 2% horse serum (16050130, Life Technologies, Merelbeke, Belgium). HPAF-II cells (CRL-1997, ATCC, Manassas, VA, USA) were maintained in DMEM supplemented with 10% FBS, and 1% penicillin-streptomycin. All cell lines were cultured at 37 °C and 5% CO_2_. For siRNA transfection experiments, 7 × 10^4^ cells were plated overnight in a 6-well plate. Cells were transfected with 100 nM of scramble siRNA (D-001810-10-05, Dharmacon, Waterbeach, UK) or a pool of 4 different siRNA targeting *Peroxiredoxin-1* gene (*PRDX1*) (L-010338-00-0005, Dharmacon, Waterbeach, UK), or *Nuclear factor erythroid 2-related factor 2* gene (*NRF2*) (L-003755-00-0005, Dharmacon, Waterbeach, UK), using the jetPRIME^®^ transfection reagent (114-15, VWR, Leuven, Belgium). One day later, the transfection medium was removed, and cells were incubated for two additional days with fresh media. After 3 days of transfection, cells were re-transfected a second time with the same abovementioned settings and kept in culture for 3 additional days (total of 6 days). Cells were then collected to assess survival or for protein extraction. In select experiments, 3 × 10^4^ cells were plated per well in a 24-well plate and let grow for 2 days. Then, cells were treated with Stattic (2798/10, Bio-Techne, Minneapolis, MN, USA) or chloroquine (C6628-25G, Sigma Aldrich, Overijse, Belgium) for 24 h. 

To generate a stable PANC-1 cell line, we used clustered regularly interspaced short palindromic repeats/CRISPR-associated protein 9 (CRISPR/Cas9) technology as described previously [20]. Briefly, two specific single-guide RNA (sgRNA) (IDT, Coralville, IA, USA) were used to knockout *PRDX1* (sgRNA1: Forward: 5′-CACCGTCAGGTATTCCTAATGCACC-3′; Reverse: 5′-AAACGGTGCATTAGGAATACCTGAC-3′) and (sgRNA2: Forward: 5′-CACCGATCTAGCATGGTAAATCTCT-3′; Reverse: 5′-AAACAGAGATTTACCATGCTAGATC-3′). The sgRNA was cloned into a GFP-expressing plasmid (48138, Addgene, Watertown, MA, USA) and then transfected into PANC-1 cells. After 24 h, GFP-positive cells were sorted by FACSAria III (BD Bioscience, Erembodegem, Belgium), and individual single cells were placed in wells of a 96-well plate (Fluorescence-activated cell sorting (FACS)-sorting settings were as described previously [21]). After three weeks of culture, clones were genotyped, and PRX-I protein ablation was confirmed by Western blot.

### 2.3. Cell Counting and Crystal Violet Staining

Cell growth-rate was measured by crystal violet (C6158, Sigma Aldrich, Overijse, Belgium). This method allows the assessment of cellular cytotoxicity following a genetic or pharmacological treatment. After treatment, living cells remain adherent to the well and can be stained with crystal violet who binds ribose molecules on DNA. The number of cells stained positively with crystal violet is proportional to the number of living cells. First, cells were fixed with a solution of 10% methanol (20847.307, VWR, Leuven, Belgium) and 10% acetic acid (84874.290, VWR, Leuven, Belgium) for 15 min at room temperature (RT). Then, the plate was dried and stained with 0.5% crystal violet solution for 15 min at RT. Finally, the plate was washed three times with tap water and let dry. The next day, absorbance was measured at 595 nm by adding 10% acetic acid solution into each well. In select experiments, cells were dissociated with trypsin (25050-014, Life Technologies, Merelbeke, Belgium), and the number of cells was estimated using the automated counter TC20^TM^ (Bio-Rad, Temse, Belgium).

### 2.4. Cell Cycle Analysis 

Cell cycle analysis can be performed using DNA-binding dyes like propidium iodide. The dye binds in proportion to the amount of DNA present in the cell. For example, cells in the S and G2 phases, which have more DNA, take up more dye and fluoresce more brightly than cells in the G1 phase. Therefore, dye’s fluorescence intensity allows the separation of cells depending on their cell cycle phase (mild fluorescence: G0/G1, moderate fluorescence: S, and high fluorescence: G2/M). To assess the cell cycle, cells were plated at 1 × 10^5^ cells/well in a 6-well plate. After 2 days, cells were dissociated with trypsin and pelleted by centrifugation at 1200 rpm for 5 min. After washing in PBS (14190094, Life Technologies, Merelbeke, Belgium), cells were fixed with 70% cold ethanol (1.00983.1000, VWR, Leuven, Belgium). After centrifugation, ethanol-fixed cells were resuspended with staining buffer (0.1% Triton X-100 (3051.3, Carl Roth, Karlsruhe, Germany), 0.2 mg/mL RNAse (19101, Qiagen, Venlo, The Netherlands), 10 µg/mL propidium iodide (PI) (556463, BD Bioscience, Erembodegem, Belgium), and 1 mM EDTA (04800683, MP Biomedical, Illkirch-Graffenstaden, France), diluted in PBS) for 30 min, at RT. The PI signal was then analyzed by FACSVerse (BD Bioscience, Erembodegem, Belgium) to assess the cell cycle. The percentage of cells in the different cell cycle phases was calculated using the Watson method.

### 2.5. Reactive Oxygen Species (ROS) Measurements

ROS measurement is usually done using the 2′,7′–dichlorofluorescin diacetate (DCFDA) fluorogenic probe. DCFDA can diffuse to the cells and become deacetylated by cellular esterase to a non-fluorescent compound, which is later oxidized by intracellular ROS into a highly fluorescent compound called 2′,7′–dichlorofluorescin (DCF). To measure ROS, first, cells were plated at 5.5 × 10^4^ cells/well in a 6-well plate. After 3 days, cells were washed with PBS and incubated for 45 min with 10 µM DCFDA probe at 37 °C (ab113851, Abcam, Cambridge, UK); similar results were obtained when cells were incubated for 25 min with 5 µM DCFDA probe at 37 °C. Then, cells were washed once with PBS, dissociated with trypsin, and pelleted by centrifugation (1200 rpm, 5 min). Finally, the pellet was resuspended in 1 mL PBS, and DCFDA fluorescence was measured by FACS. 

### 2.6. Western Blot

Cells were lysed in 50 mM Tris-base-HCl (04819638-5kg, MP Biomedical, Illkirch-Graffenstaden, France), 150 mM NaCl (0219484801, MP Biomedical, Illkirch-Graffenstaden, France), and 1% NP40 (I8896-50ML, Sigma Aldrich, Overijse, Belgium) buffer by vortexing and repetitive pipetting. Protease (11836153001, Sigma Aldrich, Overijse, Belgium) and phosphatase (4906837001, Sigma Aldrich, Overijse, Belgium) inhibitors were added just before lysis. Samples were maintained on ice during the procedure. Cell debris were pelleted by centrifugation (14,000× *g*, 10 min, at 4 °C). Proteins were quantified using a Bradford assay (23200, Thermo Fisher Scientific, Merelbeke, Belgium). Samples containing 15 to 30 µg of total proteins were electrophoresed on 7.5% to 12.5% SDS polyacrylamide gels. Polyvinylidene difluoride (PVDF) membranes (ISEQ00010, Thermo Fisher Scientific, Merelbeke, Belgium) were blocked with a solution of 5% low-fat milk diluted in Tris-buffered saline (TBS)/0.1% Tween-20 (P2287-500 mL, Sigma Aldrich, Overijse, Belgium). Membranes were incubated overnight with primary antibodies at 4 °C. The used antibodies are listed in Table 1. The next day, the membranes were washed with TBS/0.1% Tween-20 and incubated with the secondary antibodies for 1 h, at RT. After incubation, membranes were washed again and revealed using chemiluminescence (kits 34577 and 34094, Thermo Fisher Scientific, Merelbeke, Belgium). Pictures were taken with a Fusion Solo S machine (Vilber, Collégien, France). Densitometric analysis was performed using the Image Studio Lite Ver 5.2 software (LI-COR Biosciences, Lincoln, NE, USA). 

### 2.7. Immunofluorescence and Immunohistochemistry

Pancreata were fixed in 4% paraformaldehyde (PFA) (HT501128-4L, Sigma Aldrich, Overijse, Belgium) overnight at 4 °C, with gentle rotation, before paraffin embedding. Six-µm tissue sections were deparaffinized, and antigen retrieval was performed in a microwave using citrate buffer (C1909-500G, Sigma Aldrich, Overijse, Belgium) (pH 6.0). Sections were washed with PBS once and then permeabilized with PBS/0.3% Triton-100X for 5 min at RT. Sections were blocked with solution 1 (3% low-fat milk, 5% bovine serum albumin (A906-100G, Sigma Aldrich, Overijse, Belgium), 0.3% Triton-100× in PBS) for 45 min at RT. PRX-I antibody (1/200; NBP1-82558, Bio-Techne, Minneapolis, MN, USA) was diluted in solution 1 and incubated overnight at 4 °C. The next day, slides were washed with PBS/0.1% Triton-100X and incubated with secondary antibodies diluted in solution 2 (10% bovine serum albumin, 0.3% Triton-100X) at 37 °C, for 1 h. Slides were scanned using the Oyster Pannoramic 250 Flash III scanner (3DHISTECH, Budapest, Hungary). Images were selected directly from the scans. Whole-tissue quantification for PRX-I staining was performed using the HALO software (Indica Labs, Albuquerque, NM, USA). For cell labeling, cells cultured on coverslips were washed once with PBS and then fixed with 4% PFA for 30 min at RT, as described previously [22]. Then cells were permeabilized, blocked with solution 1, and labeled with Microtubule-associated protein 1A/1B-light chain (LC3) (1/100; 12741S, Cell Signaling Technology, Leiden, Netherlands) and Lysosomal-associated membrane protein 1 (LAMP1) antibodies (1/100; 1D4B, DSHB, Iowa, IA, USA) at 4 °C overnight. Pictures were taken directly from mounted coverslips using the Axiovert 200 microscope (Zeiss, Zaventem, Belgium) or the cell observer spinning disk confocal microscope (Zeiss, Zaventem, Belgium).

### 2.8. Detection of 8-Oxoguanine (8-oxoG) Adducts

8-oxoG is the most prevalent oxidative DNA damage induced by ROS that modifies the nucleic acid base guanine. To assess 8-oxoG levels using a specific antibody, cells cultured on coverslips were fixed with cold methanol for 10 min, at RT. Next, cells were treated with RNase solution (5 mg/mL in 10 mM Tris–HCl pH 7.5, 15 mM NaCl) for 1 h, at 37 °C. To denature nuclear DNA, cells were rinsed with distilled water and then dipped successively into solutions of HCl (2 N) (124630010, Life Technologies, Merelbeke, Belgium) and 10 mM Tris–HCl (pH 7.5), 15 mM NaCl, for 10 min each, at RT. Finally, cells were washed twice in PBS, prior to incubation with the primary antibody diluted in solution 1 (1/100; MAB3560, Merck, Overijse, Belgium) overnight at 4 °C. The next day, cells were incubated with a secondary antibody diluted in solution 2, and pictures were taken using a cell observer spinning disk confocal microscope (Zeiss, Zaventem, Belgium) directly from mounted coverslips.

### 2.9. Statistical Analysis

Data were presented as means ± standard error of the mean (SEM). Normality and equal variance were checked before performing the statistical analysis. Then, comparisons between the two groups were performed using an unpaired Student’s *t-*test. For all statistical analyses, the level of significance was set at *p* < 0.05. Analyses were performed using SigmaStat (version 3.1, Systat Software Inc., San Jose, CA, USA) and GraphPad Prism software (version 8, GraphPad Software, Inc., San Diego, CA, USA). * *p* < 0.05, ** *p* < 0.01, *** *p* < 0.001.

## 3. Results and Discussion

### 3.1. PRX-I Expression Is Induced during Pancreatic Carcinogenesis

To assess the expression of PRX-I in tumor tissues, we first used human transcriptomic data available at the GEPIA database (Gene Expression Profiling Interactive Analysis; http://gepia.cancer-pku.cn; accessed on 15 January 2021) to compare *PRDX1* mRNA expression in six different gastrointestinal cancers with their respective normal tissues. Among cancers related to the esophagus, stomach, liver, colon, and pancreas, the expression of *PRDX1* was only significantly increased in the latter (Figure 1A); we noted that the number of patients available in the cholangiocarcinoma cohort is limited and that a bigger number of patients is needed to confirm that *PRDX1* expression is unchanged between tumoral and normal tissues. We then focused on the pancreatic cancer cohort and separated patients into two groups based on the levels of intratumor *PRDX1* expression: *PRDX1*-High and *PRDX1*-Low (cut-off High 90% and cut-off Low 10%). The *PRDX1*-Low group exhibited a significantly better overall survival compared to the *PRDX1*-High group (Figure 1B). We found no changes in overall survival when we applied the same cut-off settings to another ubiquitously expressed gene, *ribosomal protein L24* (*RPL24*), thus highlighting the specific effect of *PRDX1* expression on patient survival (Figure 1B).

We confirmed the selective increase in PRX-I protein expression by immunohistochemistry in mouse PanIN and PDAC compared to normal pancreatic cells (Figure 1C). Whole-tissue quantification revealed that only 0.96% ± 0.33% of normal pancreas area was stained positively for PRX-I in *WT* mice, compared to 42% ± 4.67% of PRX-I-positive pancreas area in *KPE* mice (Figure 1C), indicating that almost all normal pancreatic cells do not express PRX-I and that its expression is selectively increased in pancreatic cells undergoing tumorigenesis (neoplastic or tumoral stage). Our observations are in line with a clinical report showing increased PRX-I expression in human PDAC samples [23]. Our data indicate that among different gastrointestinal cancers, PRX-I is particularly upregulated in mouse and human pancreas undergoing tumorigenesis.

### 3.2. Genetic Ablation of PRX-I Induces a Prooxidant Response Leading to Cell Cycle Blockade and Reduced Growth of Human PDAC Cells

Few functional studies have assessed the role of PRX-I in PDAC. Although a previous study found that PRX-I promotes the invasiveness of PDAC cells [17], the impact of PRX-I on their redox biology and cell growth remains unknown. To determine the role of PRX-I, we knocked-down its expression in various human PDAC cell lines using a pool of four different siRNA. A reduction of around 80% in PRX-I expression was observed in the different cell lines (Figure 2A and Appendix A). Crystal violet experiments revealed that PRX-I knockdown significantly reduced cell growth in PANC-1 (−35% ± 5.8%), MIAPaca-2 (−25% ± 3.2%), and HPAFII (−12% ± 1.8%) cells (Figure 2B,C). Among essential signaling pathways, including ERK/MAPK and JAK/STAT3, the knockdown of PRX-I only hampered the activation of STAT3, as evidenced by the reduced levels of phospho-STAT3^Y705^ (Figure 2D). Treatment of PDAC cells with Stattic, a STAT3 inhibitor, also reduced PDAC cell growth in a dose-dependent manner (Figure 2E). A recent study showed that the induction of intracellular ROS reduces STAT3 activation and promotes cytotoxic effects in PDAC cells [24]. In an independent study, we have also found that PRX-I expression correlates with STAT3 activity in the injured mouse pancreas [16]. These studies suggest that the presence of an antioxidant system, buffering ROS levels, maintains an optimal STAT3 activity. 

Our present data support the previous findings and strongly suggest that the antioxidant enzyme PRX-I promotes cell growth, at least partly, via the activation of STAT3 pathway. We then focused on the PANC-1 cell line, which has the advantage of being widely used in PDAC research. In line with previous reports [25], we first found that PRX-I expression was controlled by the transcription factor NRF2, as the knockdown of the latter resulted in reduced PRX-I protein expression (Appendix A). We then used CRISPR/Cas9 technology to generate a stable PANC1 cell line that completely lacks PRX-I protein expression (*PRDX1^KO^*) but not PRX-II (Figure 2F and Appendix A). We confirmed that PRX-I deficiency caused a 50% reduction in cell growth after six days of culture (Figure 2F). 

To test if PRX-I ablation reduced cell growth by affecting cell cycle progression, we performed FACS analysis and found that 36.0% ± 0.8% of *PRDX1^KO^* cells were in the G2/M phase compared to 15.5% ± 1.5% in *PRDX1^WT^* cells (Figure 3A). In addition, a decrease in the percentage of mitotic cells positive for phospho-histone 3 (pH3) was observed in the *PRDX1^KO^* background (Figure 3B). G2/M blockade is associated with increased levels of phospho-checkpoint kinase 2 (P-CHK-2), phospho- M-phase inducer phosphatase 3 (P-Cdc25c; a target of CHK-2), and p27^Kip1^, and can reflect the presence of excessive DNA damage [26]; all abovementioned markers were significantly upregulated in *PRDX1^KO^* compared to *PRDX1^WT^* cells (Figure 3C). Our data indicate that inhibition of PRX-I leads to G2/M blockade in PDAC cells. 

Since oxidative DNA damages can lead to G2/M blockade [27], we first measured the intracellular levels of ROS by FACS. We observed a five-fold increase in intracellular ROS production in *PRDX1^KO^* compared to *PRDX1^WT^* cells (Figure 3D), which is in line with previous research [14]. Then we measured the oxidative DNA damage marker 8-oxoguanine (8-OxoG). As expected, we found significantly higher levels of 8-OxoG in *PRDX1^KO^* compared to *PRDX1^WT^* cells (Figure 3E). Interestingly, the treatment with different doses of N-Acetylcysteine (NAC) did affect the percentage of cells in the S and G2/M phases in both *PRDX1^WT^* and *PRDX1^KO^* cells, indicating that the damages caused by PRX-I loss are not reversible by generic antioxidant treatment (Appendix A). However, we observed a reduction in the percentage of cells in the G0/G1 phase, only in the NAC-treated *PRDX1^KO^* background (Appendix A), suggesting a slight increase in cell cycle progression following NAC treatment [28]. Treatment of *PRDX1^KO^* cells with NAC resulted in a significant decrease in ROS levels, thus confirming that NAC was acting as an antioxidant compound in the used experimental conditions (Appendix A). Altogether, our results indicate that PRX-I ablation causes irreversible oxidative DNA damage, leading to disrupted cell cycle progression and reduced growth of PDAC cells. 

### 3.3. Genetic Ablation of PRX-I Alters the Autophagic Flux in PDAC Cells 

Cell cycle blockade and DNA damage could result in reduced cell survival [29]. Autophagy has been shown to promote the survival of PDAC cells, and its inhibition suppressed their growth [30]. We hypothesized that autophagy could be perturbated in cells lacking PRX-I, which may explain their reduced growth. To test this hypothesis, we measured the protein expression of essential autophagy markers. 

We found that autophagy-related gene (ATG)7 expression, which is involved in LC3 lipidation (LC3-I → LC3-II), was increased in *PRDX1^KO^* cells; in parallel, the expression of ATG5-ATG12 and ATG16L1 involved in autophagosome elongation was decreased (Figure 4A). This first results highlight the presence of changes in basal autophagic activity. To confirm autophagy disruption, we assessed the levels of LC3 and p62 proteins; the latter is degraded when autophagy is active [31], and the accumulation of both markers reflects the presence of defective autophagy. As expected, we found an increase in LC3-II and p62 content in *PRDX1^KO^* cells (Figure 4A). In addition, LC3 immunofluorescence revealed that some of the cells lacking PRX-I exhibited two-fold larger LC3 dotes compared to *PRDX1^WT^* cells (Figure 4B–D; white arrows); these larger LC3 dots, which are indicative of defective autophagy [31], did not colocalize with LAMP1, a marker of lysosomal membranes, emphasizing a reduced formation of autophagolysosomes. Treatment of *PRDX1^WT^* cells with chloroquine reduced their survival in a dose-dependent manner (Figure 4E), thus confirming that autophagy is essential to maintain the survival of PDAC cells. However, the combination of PRX-I ablation and chloroquine resulted in an additive rather than the synergetic response in *PRDX1^KO^* cells (Figure 4E). Our data indicate that the removal of PRX-I disrupts the basal autophagic flux present in pancreatic cancer cells. Signaling pathways responsible for autophagy perturbation in PRX-I-deficient cells are complex and multiple. Since we demonstrated that PRX-I promotes STAT3 activation, we wondered if STAT3 is responsible for disrupted autophagy observed in cells lacking PRX-I. Although pharmacological inhibition of STAT3 by Stattic induced a significant increase in ROS levels (Appendix A), it did not alter the autophagic flux in PDAC cells. Namely, there was no change in the diameter of LC3 dots and in LC3 total protein content between Stattic-treated and untreated PANC-1 cells (Appendix A). These experiments indicate that PRX-I loss alters autophagy in PDAC cells independently on the STAT3 pathway. Besides STAT3, AMP-activated protein kinase (AMPK) has been described as an activator of autophagy [32,33]; we have found that the phosphorylation of Acetyl-CoA carboxylase (ACC), a direct substrate of AMPK [34], was significantly decreased in *PRDX1^KO^* compared with *PRDX1^WT^* cells (data not shown), highlighting a reduction in AMPK downstream activity. Therefore, it seems that loss of PRX-I can impact autophagy, at least partly, through a modulation in AMPK activity. However, a complete elucidation of the underpinning mechanisms warrants further investigation. 

In summary, we found that PRX-I is increased during pancreas tumorigenesis and that its ablation in human PDAC cells compromises their growth. Mechanistically, the ablation of PRX-I led to increased intracellular ROS generation and oxidative DNA damage, probably leading to cell cycle blockade. Additionally, the lack of PRX-I affected important biological pathways essential for PDAC cell survival, including STAT3 and autophagy. Specific inhibitors for PRX-I have been developed recently [35,36], their validation in the context of PDAC will deserve future in vitro and in vivo investigations.

## 4. Conclusions

In this paper, we report for the first time a pro-oncogenic role for PRX-I in PDAC. Using molecular and redox biology approaches, we provide evidence that PRX-I impacts a multitude of pathways related to cell cycle progression, autophagy, and survival, through buffering the intracellular levels of ROS and the resulting oxidative DNA damages. It is important to emphasize the selective expression of PRX-I in PDAC cells as compared to normal pancreatic cells, making PRX-I a druggable oncogenic target with few expected side effects on normal tissues. This selectivity has been partly confirmed in a recent study on mice [16]; specifically, in a model of experimental acute pancreatitis, ablation of PRX-I in the pancreas with mild inflammation and low PRX-I expression showed no impact on pancreatic cells, whereas PRX-I inactivation in the pancreas with severe inflammation and high PRX-I expression exhibited protective effects on pancreatic cells [16]. Therefore, PRX-I could be considered as a specific target in tumors or injured cells (e.g., exposed to inflammatory stimuli and/or oncogenic mutations). Our previous findings [16] and those described in the present work encourage the implementation of prospective studies to test specific pharmacological inhibitors for PRX-I in mice models of pancreas tumorigenesis. 

## Figures and Tables

**Figure 1 antioxidants-10-00570-f001:**
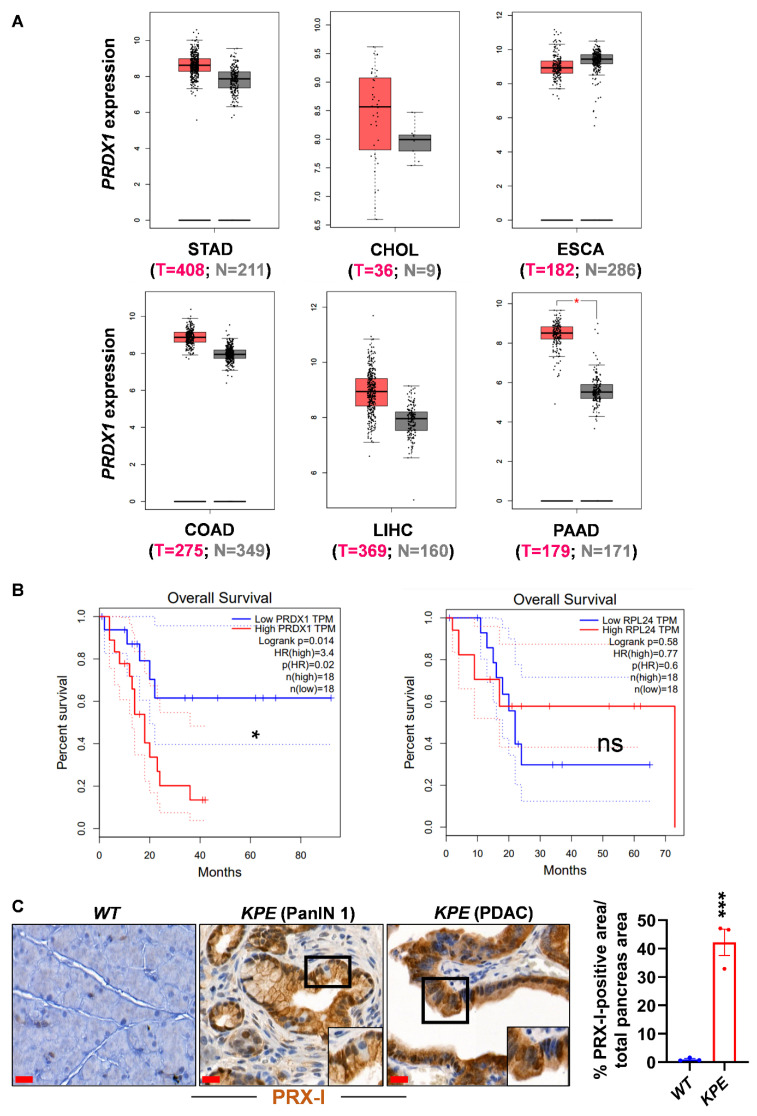
Peroxiredoxin-I (PRX-I) expression is upregulated in pancreatic tumors. (**A**) *Peroxiredoxin-1* gene (*PRDX1*) expression in the indicated gastrointestinal tumors (pink) and their corresponding normal tissues (grey) available from the GEPIA database. Red asterisk indicates statistical significance. Ordinate scale: Log_2_ (TPM + 1) (**B**) Survival curve for *PRDX1*-Low (blue) and *PRDX1*-High (Red) pancreatic cancer cohort from the Gene Expression Profiling Interactive Analysis (GEPIA) database. Black asterisk indicates statistical significance. ns: not significant. (**C**) Representative images of Peroxiredoxin-I immunostaining on pancreas from *wild-type* (*WT*) mice untreated with cerulein (left panel) and *ElastaseCreER-LSLKras^G12D^-p53^R172H^* (*KPE*) mice treated with cerulein for six weeks, followed by 14 weeks of recovery to allow tumor formation (middle and right panels). Pancreas sections from *KPE* mice contain pancreatic intraepithelial neoplasia (PanIN) and pancreatic ductal adenocarcinoma (PDAC). Bars: 20µm. Whole-tissue quantification of Peroxiredoxin-I protein expression from *WT* (*n* = 3) and *KPE* (*n* = 3) mice is presented in the bar graph. Data are mean ± SEM. Statistical significance was tested by Student’s *t-*test (* *p*<0.05; *** *p* < 0.001). STAD, stomach adenocarcinoma; CHOL, cholangiocarcinoma; ESCA, esophageal carcinoma; COAD, colon adenocarcinoma; LIHC, liver hepatocellular carcinoma; PAAD, pancreatic adenocarcinoma.

**Figure 2 antioxidants-10-00570-f002:**
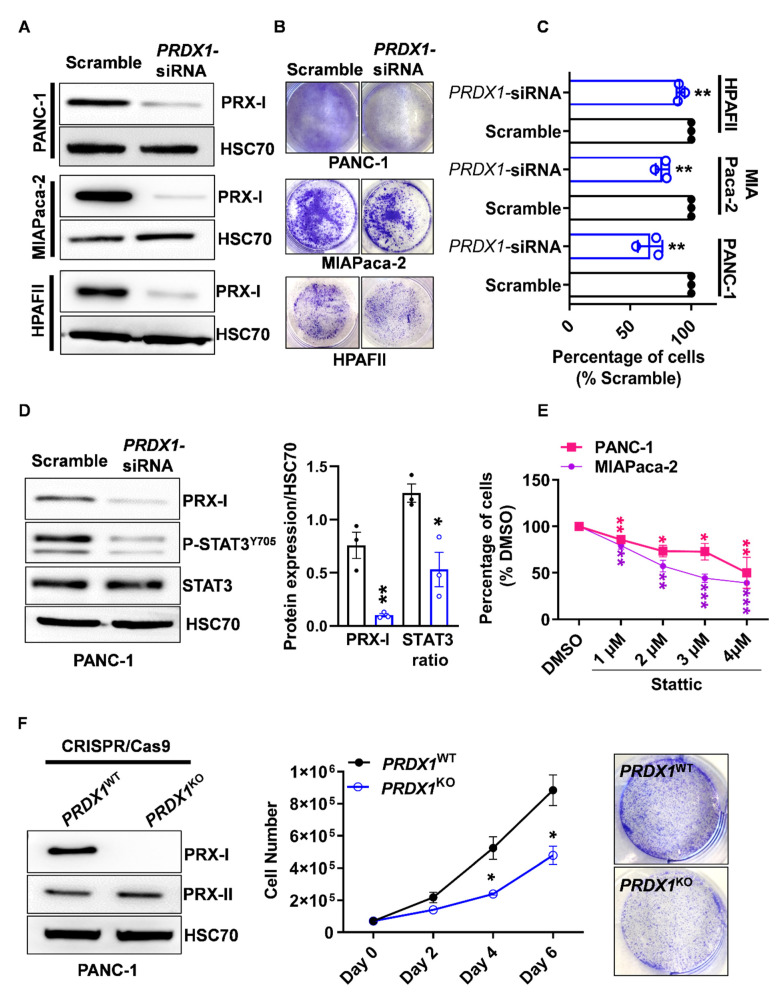
Peroxiredoxin-I (PRX-I) ablation reduces the growth of human pancreatic ductal adenocarcinoma (PDAC) cells. (**A**) Representative Western blot results for PRX-I on multiple human PDAC cell lines transfected for six days with scramble or specific *Peroxiredoxin-1* gene (*PRDX1*)*-*siRNA. Heat-shock cognate 70 (HSC70) was used as a loading control. On each cell line, siRNA transfection was performed on three independent culture passages (*n* = 3) (**B**) Representative crystal violet staining for PDAC cell lines after six days of transfection with scramble or specific *PRDX1-*siRNA. (**C**) Determination of cell percentages from experiments shown in panel B (*n* = 3). The percentage in the scramble conditions was considered to be 100%. (**D**) Western blot on PANC-1 cells after transfection with scramble or specific *PRDX1-*siRNA; Western blots were performed on three independent culture passages (*n* = 3). The corresponding densitometric quantification for blots in Panel D is also available (right panel) (*n* = 3). The signal transducer and activator of transcription 3 (STAT3) ratio is Phospho-STAT3^Y705^/STAT3 normalized to HSC70. (**E**) Determination of cell percentages in PANC-1 and MIAPaca-2 cell lines, by crystal violet staining, after 24 h of DMSO (0.04%, considered 100%) or Stattic treatment; Pharmacological treatment was performed on four independent culture passages (*n* = 4). (**F**) Western blot on stable PANC-1 cell lines expressing (*PRDX1^WT^*) or completely lacking (*PRDX1^KO^*) PRX-I expression; experiments were performed on at least three independent culture passages (*n* = 3). Quantification of *PRDX1^WT^* and *PRDX1^KO^* cells was determined at different time points using an automated cell counter; counting was performed on three independent culture passages (*n* = 3). Representative pictures of crystal violet staining showing a lower number of cells after six days of culture in the *PRDX1^KO^* line. Data are mean ± SEM. Statistical significance was tested by Student *t-*test (* *p* < 0.05, ** *p* < 0.01, *** *p* < 0.001).

**Figure 3 antioxidants-10-00570-f003:**
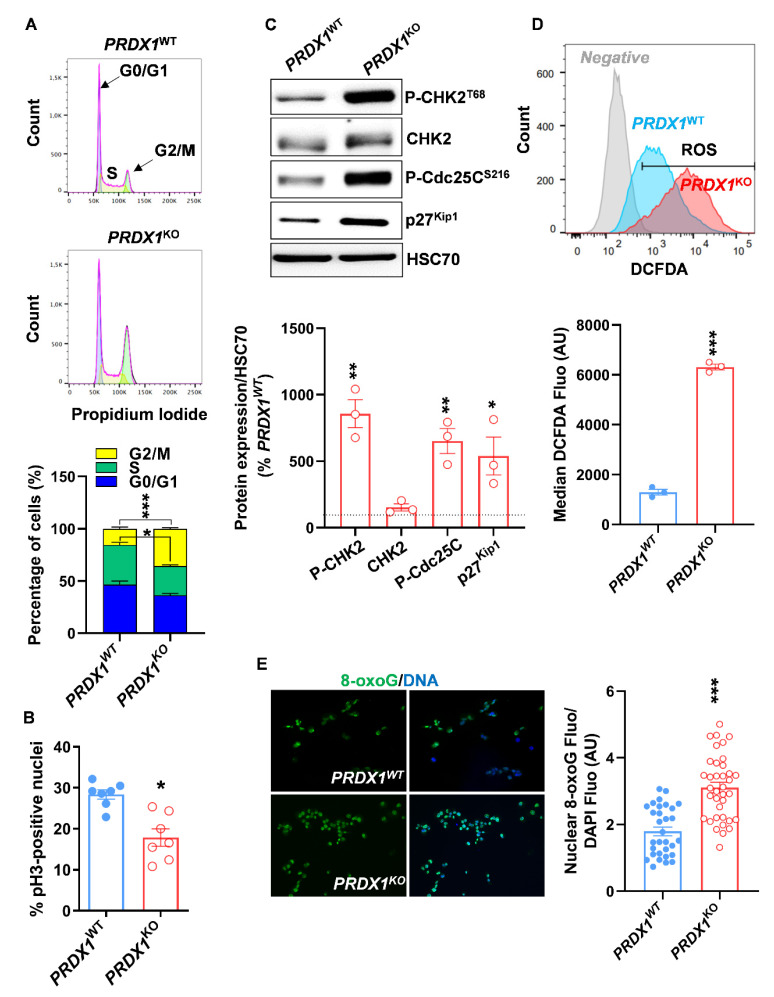
Peroxiredoxin-I (PRX-I) ablation causes oxidative DNA damages and cell cycle blockade in human pancreatic ductal adenocarcinoma (PDAC) cells. (**A**) Typical plots from fluorescence-activated cell sorting (FACS) analysis of cell cycle in *PRDX1^WT^* and *PRDX1^KO^* cells and the corresponding quantification; cell cycle analysis was performed on four independent culture passages (*n* = 4). (**B**) Percentage of phospho-histone 3 (pH3)-positive nuclei quantified on immunolabeled *PRDX1^WT^* and *PRDX1^KO^* cells; immunolabeling was performed on three independent culture passages (*n* = 3), and seven fields were randomly selected from the three experiments for pH 3 quantification. (**C**) Western blot for DNA damage markers and their corresponding densitometric quantification. Heat-shock cognate 70 (HSC70) was used as a loading control; blots were performed on three independent culture passages (*n* = 3). The dashed line, set at 100%, represents the expression level in *PRDX1^WT^* cells. (**D**) Typical plot for FACS analysis of DCFDA fluorescence (Reactive Oxygen Species, ROS levels) in *PRDX1^WT^* and *PRDX1^KO^* cells and the corresponding quantification. AU: Arbitrary Units. Measurements were performed on three independent culture passages using 10 µM of DCFDA (*n* = 3) or 5µM DCFDA (*n* = 3). Both DCFDA doses (5 and 10 µM) gave similar results. (**E**) Representative images of 8-oxoguanine (8-oxoG) nuclear immunolabeling in *PRDX1^WT^* and *PRDX1^KO^* cells; experiments were performed on two independent culture passages (*n* = 2), and nuclear 8-oxoG fluorescence intensity was quantified in 31-to-37 nuclei selected randomly. Specific nuclear 8-oxoG fluorescence was normalized to DAPI fluorescence. Data are mean ± SEM. Statistical significance was tested by Student’s *t-*test (* *p* < 0.05, ** *p* < 0.01, *** *p* < 0.001).

**Figure 4 antioxidants-10-00570-f004:**
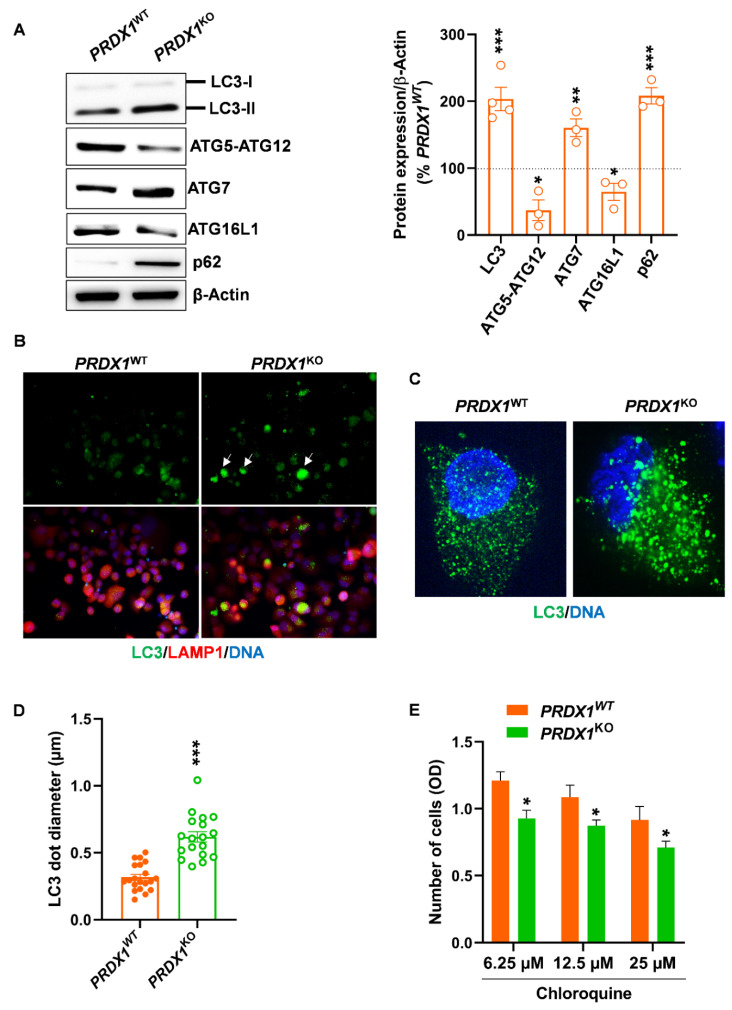
Peroxiredoxin-I (PRX-I) ablation disrupts basal autophagic flux in human pancreatic ductal adenocarcinoma (PDAC) cells. (**A**) Western blot for autophagy markers and their corresponding densitometric quantification. β-Actin was used as a loading control; Western blots were performed on three-to-four independent culture passages (*n* = 3–4). The dashed line, set at 100%, represents the expression level in *PRDX1^WT^* cells. (**B**) Representative pictures of Microtubule-associated protein 1A/1B-light chain (LC3) and Lysosomal-associated membrane protein 1 (LAMP1) immunolabeling on *PRDX1^WT^* and *PRDX1^KO^* cells; immunolabeling was performed on three independent culture passages (*n* = 3). White arrows show autophagy-defective cells with large LC3 dots. (**C**) Representative confocal pictures of LC3 immunolabeling in one *PRDX1^WT^* cell and one *PRDX1^KO^* cell. (**D**) LC3 dot diameter measured from confocal pictures, as shown in panel C; LC3 immunolabeling was performed on three independent cultures (*n* = 3), and 18-to-20 cells were randomly selected from confocal images to quantify LC3 dot diameter. (**E**) Cell number measured after 24 h of chloroquine treatment of *PRDX1^WT^* and *PRDX1^KO^* cells; pharmacological treatment was performed on three independent culture passages (*n* = 3). Data are mean ± SEM. Statistical significance was tested by Student’s *t-*test (* *p* < 0.05, ** *p* < 0.01, *** *p* < 0.001).

**Table 1 antioxidants-10-00570-t001:** Antibodies used in Western blot experiments. SCB: Santa Cruz Biotechnology, CST: Cell Signaling Technology, SA: Sigma Aldrich, BSA: bovine serum albumin.

Antibody	Reference	Dilution	Conditions
PRX-I	NBP1-82558, BioTechne Minneapolis, MN, USA	1/1000	5% Milk, overnight, 4 °C
PRX-II	Sc-515428, SCB, Heidelberg, Germany	1/500	5% Milk, overnight, 4 °C
P-STAT3^Y705^	9145S, CST, Leiden, The Netherlands	1/1000	5% BSA, overnight, 4 °C
STAT3	12640S, CST, Leiden, The Netherlands	1/1000	5% BSA, overnight, 4 °C
P-CHK2^T68^	2197S, CST, Leiden, The Netherlands	1/1000	5% BSA, overnight, 4 °C
CHK2	2662S, CST, Leiden, The Netherlands	1/1000	5% BSA, overnight, 4 °C
P-Cdc25C^S216^	4901S, CST, Leiden, The Netherlands	1/1000	5% BSA, overnight, 4 °C
p27^Kip1^	3686S, CST, Leiden, The Netherlands	1/1000	5% BSA, overnight, 4 °C
LC3	12741S, CST, Leiden, The Netherlands	1/1000	5% BSA, overnight, 4 °C
ATG5-ATG12	4180T, CST, Leiden, The Netherlands	1/1000	5% BSA, overnight, 4 °C
ATG7	8558T, CST, Leiden, The Netherlands	1/1000	5% BSA, overnight, 4 °C
ATG16L1	8089T, CST, Leiden, The Netherlands	1/1000	5% BSA, overnight, 4 °C
p62	Sc-28359, SCB, Heidelberg, Germany	1/500	5% Milk, overnight, 4 °C
HSC70	Sc-7298, SCB, Heidelberg, Germany	1/5000	5% Milk, overnight, 4 °C
β-Actin	A5441-100UL, SA, Overijse, Belgium	1/5000	5% Milk, overnight, 4 °C

## Data Availability

Not applicable.

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
