# Peer review of "Genetic Inactivation of Peroxiredoxin-I Impairs the Growth of Human Pancreatic Cancer Cells"

_antioxidants, 2021, doi:10.3390/antiox10040570_

Round 1

Reviewer 1 Report

The manuscript entitled “Genetic inactivation of Peroxiredoxin-I impairs the growth of human pancreatic cancer cells”, submitted by Hajar Dajou and colleagues, is focused on the role of PRX-1 in Pancreatic ductal adenocarcinoma. The authors demonstrate that PRX-1 ablation increases oxidative stress and DNA damage and reduces autophagic flux. This exciting work fits with the aim of the journal, and the results support the initial hypothesis. In addition, the experiments are well-controlled and clearly presented. I would be happy to recommend the publication of the manuscript after consideration of the points below:

Major concerns:

  • I consider that the concentration and incubation time of DCFDA dye is excessive. The concentration customarily used is 5 µM for 20-30 minutes. I encourage you to do the same experiments using these conditions to confirm the results presented here.
  • The inhibition of STAT3 is only employed to see cell proliferation. I suggest using the inhibitor to validate the effect on oxidative stress and autophagy.
  • The effect of NAC as an antioxidant is dose-dependent and different for each cell type. A confirmation that this antioxidant decreases DCFDA levels in Fig Sup 2 is required.
  • Since the ablation of PRX-1 impacts autophagy, the restoration of autophagic flux using an activator of autophagy would improve the significance of the results.

Minor concerns:

  • Double spaces in line 38 and 255.
  • The manuscript indicates that the PANC-1 cell line is more sensitive to PRX-1 reduction than the other two. However, it seems that the effect in PANC-1 and MIAPaca-2 is similar. Even the impact of stattic is higher in the MIAPaca-2. I recommend removing this statement in line 204.

Author Response

Dear reviewer,

Thank you for your constructive comments and suggestions that improved the quality of our manuscript. Please see the attached PDF file, which contains a complete point-by-point reply and some additional results.

Reviewer 2 Report

Manuscript ID: antioxidants-1159110

Title: Genetic inactivation of Peroxiredoxin-I impairs the growth of 2 human pancreatic cancer cells

Journal: Antioxidants

Authors: Dahou et al.

The above manuscript is very interesting. The purpose of the study was to analyze the effect of peroxiredoxin-1 inactivation on the growth of human pancreatic acinar cells. The authors found that peroxiredoxin-1 ablation led to reduced survival of pancreatic ductal adenocarcinoma cells. This effect was mainly due to an increase in the production of reactive oxygen species (ROS), accumulation of oxidative DNA damage (8-oxoG), and cell cycle blockade at G2/M. Moreover, ablation of peroxiredoxin-1 disrupted the autophagic flux in those cells.

On the other hand, it must be pointed out that the manuscript contains some deficiencies and errors that should be corrected before the article could be accepted for publication.

List of deficiencies and errors:

  1. Abstract, lines 11-13. The authors stated that: “we first made the intriguing discovery that Peroxiredoxin-I gene (PRDX1) 1 expression is selectively increased in human pancreatic tumors, but not in other gastrointestinal cancers”. This statement is not correct. Figure 1A in the manuscript indicates that peroxiredoxin-I gene expression is increased in almost all gastrointestinal cancers. The difference between pancreatic adenocarcinoma and other neoplasm in the gut is that the increase in PRDX1 expression in pancreatic adenocarcinoma is statistically significant in comparison to the expression of this in normal tissue (data from GEPIA database). For this reason, the authors should change their statement according to the analyzed data.
  2. Page 1, Introduction, line 41. The authors should check the units of catalytic efficiency.
  3. Page 1 introduction. The authors should write in the introduction that pancreatic function are regulated by nervous and endocrine pathways, and cholecystokinin, and secretin play a particular role in these mechanisms (PMID: 25716961). Cholecystokinin is also credited with participating in the processes leading to the development of pancreatic cancer (PMID: 2285582, PMID: 24177032).
  4. Pages 2-4. Materials and Methods. For each chemical, equipment and software, the authors should provide the manufacturer’s name, city and country.
  5. Page 2, Material and Methods, section 2.1. Mice and ethical approval”. The authors should provide the total number of animals used, age, body mass as well as their gender. The authors In the case of animals of different sexes, the authors should write whether the females and males were kept in separate cages or the animals of both sexes were in shared cages. Pregnancy may affect the results of the study.
  6. Page 2, Material and Methods, section “2.1. Mice and ethical approval”. The authors stated that: “ElastaseCreER- 54 LSLKrasG12D-p53R175H mice [2], maintained in a CD1-enriched background, were treated with cerulein (125µg/Kg; 24252, Eurogentec) for 4 weeks to induce pancreas inflammation, and then kept for 14 weeks to allow tumor development”. The authors should present the source of these animals, as well as should provide more detail on cerulein administration. Was the dose of 125µg/kg a daily dose or a single administration dose? How many doses were administered daily? Was cerulein administered daily for 4 weeks or only on certain days? What was the route of cerulein administration and what volume was of each dose? How was the euthanasia of animals carried out? In addition, the authors should state whether all are animals were treated with cerulein or whether there was any control group? Are the images shown in Fig.1C from different arts f the same pancreas, or is the image of the normal pancreas from the control group without cerulein administration?
  7. All abbreviations should be presented in their full name in the place where they appear for the first time.
  8. Figures and their legends must be understandable without references to the body of the manuscript. For this reason, all abbreviations should be presented in full name in Figure legends.
  9. Page 2, Material and methods, section “Cell lines, siRNA transfections and pharmacological treatments”. The authors should provide more details on cell line used in the study, including the source of these cells. In addition, the main principle of the medicine is ”primum non nocere”. For this reason, the authors must investigate the effect of peroxiredoxin-I (PRX-1) ablation on the survival of non-neoplastic cells. The lack of negative effects of PRX-1 ablation on non-neoplastic would indicate a possible usefulness of this method in the treatment of pancreatic cancer. In contrast, decreased survival of non-neoplastic cells and/or symptoms of severe damage of these cells would indicate that the author’s concept is useless in the clinical therapy of pancreatic cancer.
  10. Pages 2-3, Material and methods, sections 2.3; 2.4; 2.5, 2.8. The authors should present the idea of methods of performed determinations.
  11. Page 4, Results and Discussion, lines 156-168 and Figure 1 A and B. This part of the manuscript is based on data obtained from the GEPIA database and for this reason it should presented in the Introduction not in the Results and Discussion.
  12. Figure 2. Figure 2A probably shows a representative effect of transfection with specific PRDX1-siRNA on PRX-1 expression in different human pancreatic ductal adenocarcinoma cell lines. However, the authors should also provide number of observation and the graph showing densitometric analysis of these effects.
  13. Figure legend 2C, D, E F (panel 2). What does n=3 or n=4 mean? Is it the number of measurements performed on the same group of cells or the number of separate sets of cells subject to measurement?  The authors should provide clearly statement in this matter. The same problem is in other figures.
  14. Figure 2F (panel 1). The authors should also provide number of observation and show the graph showing densitometric analysis of presented effects.

Author Response

(The authors gave the same response as above.)

Reviewer 3 Report

The study “Genetic inactivation of Peroxiredoxin-I impairs the growth of human pancreatic cancer cells” aimed to investigate a pro-oncogenic role of peroxiredoxin-I (PRX-I) in pancreatic ductal adenocarcinoma (PDAC). In my opinion, the topic is interesting and noteworthy. The Authors indicated that: i) PRX-I expression is upregulated in pancreatic tumors; ii) genetic ablation of PRX-I induces cell cycle blockade, reduced growth and alteration of the autophagic flux in human PDAC cells. The research is well designed, the methods are described clearly and the results are presented expertly. I have a few suggestions and questions that Authors should consider to improve their manuscript.

  1. The epidemiology data related PDAC (incidence and mortality) can be included in the Introduction.
  2. Line 102: “Cells were plated at 55 x 103 cells/well…” – this determination of the number of cells raises doubts: 5.5 x 103 or 5.5 x 104 ?
  3. Figure 1A: Data regarding cholangiocarcinoma are limited (T=36, N=9). A comment on this limitation should be included in the text.
  4. Figure 1C: Representative microscopic images were presented. Did the Authors perform a quantitative analysis of PRX-I protein expression in the mouse pancreas sections containing PanIN and PDAC as well as normal sample?
  5. Figure 2C: how many wells were analyzed during each of the three independent experiments?
  6. Figure 2D: legend to bar graph should be added.
  7. Figure 4A: the order of representative blots does not correspond to the order in the bar graph.
  8. How many representative images were analysed in each of the independent experiments?

I recommend the study for publication after minor revision.

Author Response

(The authors gave the same response as above.)

Round 2

Reviewer 1 Report

The authors have address all my concerns. Nice work

Reviewer 2 Report

In the opinion of the reviewer, the current version of the manuscript is ready for printing.